# Personalized Management of Sudden Death Risk in Primary Cardiomyopathies: From Clinical Evaluation and Multimodality Imaging to Ablation and Cardioverter-Defibrillator Implant

**DOI:** 10.3390/jpm13050877

**Published:** 2023-05-22

**Authors:** Davide Lazzeroni, Antonio Crocamo, Valentina Ziveri, Maria Francesca Notarangelo, Davide Rizzello, Matteo Spoladori, Davide Donelli, Giovanna Cacciola, Diego Ardissino, Giampaolo Niccoli, Giovanni Peretto

**Affiliations:** 1Prevention and Rehabilitation Unit of Parma, IRCCS Fondazione Don Gnocchi, 43100 Parma, Italy; davide.lazzeroni@gmail.com (D.L.);; 2U.O.C. di Cardiologia, Azienda Ospedaliero-Universitaria di Parma, 43100 Parma, Italy; 3Department of Cardiac Electrophysiology and Arrhythmology, IRCCS San Raffaele Scientific Institute, 20132 Milan, Italy; peretto.giovanni@hsr.it

**Keywords:** cardiomyopathies, sudden death, hypertrophic cardiomyopathy, dilated cardiomyopathy, arrhythmogenic cardiomyopathy, personalized medicine, ventricular arrhythmias, ventricular arrhythmia ablation, cardioverter-defibrillator, genetics

## Abstract

Sudden cardiac death represents the leading cause of death worldwide; although the majority of sudden deaths occur in an elderly population with coronary artery disease, some occur in young and otherwise healthy individuals, as is the case of cardiomyopathies. The aim of the present review is to provide a stepwise hierarchical approach for the global sudden death risk estimation in primary cardiomyopathies. Each individual risk factor is analyzed for its contribution to the overall risk of sudden death for each specific cardiomyopathy as well as across all primary myocardial diseases. This stepwise hierarchical and personalized approach starts from the clinical evaluation, subsequently passes through the role of electrocardiographic monitoring and multimodality imaging, and finally concludes with genetic evaluation and electro-anatomical mapping. In fact, the sudden cardiac death risk assessment in cardiomyopathies depends on a multiparametric approach. Moreover, current indications for ventricular arrhythmia ablation and defibrillator implantation are discussed.

## 1. Introduction

### 1.1. Classification of Cardiomyopathies from a Traditional Viewpoint to a Personalized Approach

For more than two millennia, physicians have tended towards personalization since the days of Hippocratic medicine. Similarly, the history of medicine in the field of cardiomyopathies (CMPs), hereditary genetic heart diseases, has been characterized by a constant process of personalization, and CMP and its arrhythmic risk remain highly tailored today, in contrast to the less personalized management of common heart diseases (such as ischemic heart disease). The history of CMPs (Figure 1), as inherited genetic disorders, dates back to 1902 when the first link between genetic inheritance and disease susceptibility was proposed. The first description of cardiac hypertrophy of unknown origin, hereditary and associated with sudden death, dates back to the middle of the last century. Only in 1990 was the genetic origin of hypertrophic cardiomyopathy (HCM) demonstrated [1,2]. Finally, the sequencing of the entire human genome in 2003 brought a huge boost in understanding the genetic background of diseases. Nowadays, medical science is moving beyond the border of the genome towards new horizons (proteome, metabolome, and epigenome) in order to bridge the gap between genotype and phenotype [3,4]. Personalized medicine (PM) was defined by Horizon 2020 Advisory Group as “*a medical model using characterization of individuals’ phenotypes and genotypes for tailoring the right therapeutic strategy for the right person at the right time, and/or to determine the predisposition to disease and/or to deliver timely and targeted prevention*” [5]. Similarly, the history CPM classification is moving from an exclusively phenotypic classification approach to a PM approach, thereby integrating genotype and phenotype information (Figure 1).

In 2006, the American Heart Association (AHA) defined and classified cardiomyopathies as “a heterogeneous group of diseases of the myocardium associated with mechanical and/or electrical dysfunction, which usually (but not invariably) exhibit inappropriate ventricular hypertrophy or dilatation and are due to a variety of causes that frequently are genetic” [6].

Two years later, in 2008, the European Society of Cardiology Working Group on Myocardial and Pericardial Diseases proposed a new classification of cardiomyopathies that were defined as “*myocardial disorders in which the heart muscle is structurally and functionally abnormal, and in which coronary artery disease, hypertension, valvular and congenital heart disease are absent or do not sufficiently explain the observed myocardial abnormality*” [7].

Five types of cardiomyopathies are recognized according to their morpho-functional phenotype:(1)HCM;(2)Dilated cardiomyopathy (DCM);(3)Arrhythmogenic cardiomyopathy (ACM);(4)Restrictive cardiomyopathy;(5)Unclassified cardiomyopathy.

Each phenotype is then sub-classified into familial/genetic and non-familial/non-genetic forms.

More recently, in 2013, the World Heart Federation supported a new nosology system termed “MOGE(S)”, which differentiates cardiomyopathies according to phenotype description, extra-cardiac involvement, transmission pattern, and genotype.

MOGES classification describes cardiomyopathy with five attributes:M: morpho-functional phenotype;O: involved organs;G: genetic/familial disease or not familial;E: etiology (genetic or not);S: functional status: stage of the disease according to NYHA class or AHA/ACC classes for heart failure [8] (Figure 2).

### 1.2. Sudden Cardiac Death Risk in Cardiomyopathies: From a Traditional Viewpoint to a Personalized Approach

Sudden cardiac death (SCD) is defined as a sudden natural death presumed to be of a cardiac cause that occurs within 1 h of onset of symptoms in witnessed cases and within 24 h of last being seen alive when it is unwitnessed [9].

Sudden death (SD) occurs mostly due to cardiovascular disease and accounts for approximately 50% of all cardiovascular deaths, being the first manifestation of cardiac disease in about 50% of cases. While the majority of cardiac arrests occur in an elderly population with CAD (which accounts for up to 75–80% of SCD cases), some occur in young and otherwise healthy individuals, as is the case of CMPs [10].

Although the indication for ICD implantation in CAD remains strongly dependent on a single parameter, namely the left ventricular ejection fraction (LVEF) < 35%, in the universe of CMPs, SCD risk depends on several variables and requires a multiparametric and personalized evaluation, resulting therefore in a less “LVEF-centric” approach in which a pivotal role is still represented by the clinical and family history evaluations and in which genetics is showing an increasingly promising role.

In fact, several risk stratification schemes and calculators have been developed for inheritable CMPs, such as HCM, ACM, and lamin A/C (LMNA) cardiomyopathy [11].

Regarding HCM, in 2020, US guidelines endorsed a single risk factor-based decision model [12].

On the other hand, in 2014, the European Society of Cardiology (ESC) guidelines abandoned risk assessment strategies focused on counting or weighting single risk factors and adopted a scoring system that integrates binary and continuous variables to predict the risk of SCD over 5 years [13]. Similarly, the latest 2022 ESC Guidelines for the management of patients with ventricular arrhythmias and the prevention of sudden cardiac death recommended a multiparametric approach (including genetics) in SCD risk stratification in ACM and DCM [10].

In conclusion, the SCD risk assessment in CMPs is thus moving towards a “personalized approach” in order to progressively overturn the historical “*LVEF-centric view*” in favor of a hierarchical approach which starts from the clinical evaluation and extends to a targeted and integrated use of genetics, passing through multimodal imaging. This hierarchical and multiparametric approach is the subject of the present review.

## 2. The Hierarchical and Multiparametric Approach of SCD Risk Estimation in Cardiomyopathies

Although cardiomyopathies may present several different phenotypes as well as many different genotypes (often overlapping each other), an overlap of SCD risk factors is also present among all the different forms of CMPs; therefore, in the present review, after examining each risk factor of SCD in primary CMPs, a hierarchical approach for the global estimation of SCD risk is proposed. This stepwise hierarchical approach starts from the clinical evaluation, an essential and highly informative time, subsequently passing through the role of electrocardiographic monitoring (both 24 h and during exercise test) and multimodality imaging and finally concluding with genetic evaluation. As a result, risk stratification is defined as an integrated assessment of several variables obtained through different tests, which can be repeated over time and subject to periodic risk reassessment. The present stepwise hierarchical approach is shown in Figure 3.

### 2.1. Step 1: Clinical Evaluation and Family History (of CMPs and SCD)

From a PM viewpoint, the clinical evaluation represents the first and most relevant step in CMP diagnosis, differential diagnosis, and prognostication. One of the most useful and simple parameters that should be considered in SCD risk evaluation is the age of patients and the age at the presentation of the disease (onset).

Hypertrophic cardiomyopathy is the second most common cardiomyopathy occurring during pediatric age with reported annual SCD mortality rates ranging from 1% to 2.5%, thus being considered the most common cause of death [14,15]. Conversely, in patients with HCM > 60 years of age, SCD has a low incidence [14]. Although not considered a traditional marker of SCD, pediatric onset or young-adult presentation should be considered as a crucial factor in SCD stratification.

Moreover, family history of juvenile SCD represents one of the most powerful predictors of SCD in all CMPs, being considered in SCD risk stratification of HCM, DCM, and ACM [10].

The evaluation of symptoms allows for the assessment of two other important predictors of SCD in all CMPs: NYHA Classification for heart failure and unexplained syncope (defined as unexplained loss of consciousness within the previous 6 months in the absence of vagal prodrome), the latter having the greatest impact on SCD risk when compared to all other known predictors. Moreover, angina or palpitations, common in all CMPs, should lead to further investigations in order to assess myocardial ischemia and arrhythmic risk.

Previous medical history is also required to rule out common causes of ventricular dysfunction (such as hypertension, myocardial ischemia, valve dysfunction, and prior exposure to toxins and environmental pathogens) in order to suspect and diagnose a genetic myocardial disease [16,17]. Furthermore, multiorgan involvement evaluation is crucial not only for diagnosing HCM phenocopies but also for SCD risk stratification, since systemic diseases (e.g., Anderson–Fabry or systemic amyloidosis) are associated with a significantly worse prognosis than primitive cardiac diseases [16] (Table 1).

Similarly, the definition of different inheritance patterns represents a critical moment during clinical evaluation, since the family tree and family histories within it provide fundamental diagnostic and, consequently, prognostic data.

### 2.2. Step 2: ECG and Echocardiography

Even in the era of great technologies, the electrocardiogram represents the first and irreplaceable tool for the diagnosis and prognosis of all CMPs. Firstly, finding a normal ECG in CMPs is extremely rare (high negative predictive value); secondly, abnormal ECG may be the only or the first phenotypic manifestation of CMPs. Finally, ECG, especially when combined with data from other tests, provides suspicion of a specific underlying disease (e.g., cardiac amyloidosis) whose diagnosis is of crucial importance in estimating the risk of sudden death, since phenocopies are characterized by an adverse prognosis [16]. The most peculiar clinical, ECG, echocardiographic, and CMR features useful in the differential diagnosis between different forms of left ventricular hypertrophy from the hypertensive heart and athlete’s heart to HCM and HCM phenocopies (Anderson–Fabry disease, familial amyloidosis, Danon disease, mitochondrial CMP) are shown in Table 1. Among these aspects, multiorgan involvement primarily characterizes HCM phenocopies. Additionally, the following ECG anomalies are useful in the differential diagnosis of HCM phenocopies: low QRS voltages are specific for familial amyloidosis; AV blocks can be more common in Anderson–Fabry disease, familial amyloidosis, and Danon disease; and finally, ST and T anomalies are typical of HCM. The main echocardiographic and CMR differences will be described below.

Echocardiography represents the first line and the gold standard exam to suspect CMPs and is able to identify each phenotype when manifested, typical of all CMPs, especially in HCM and DCM. Echocardiography is recommended in all CMP patients and should be performed every 1–2 years in clinically stable patients.

#### 2.2.1. Hypertrophic Cardiomyopathy

In HCM, the following echocardiographic parameters represent important prognostic markers of SCD risk:(1)Left ventricle wall thickness (LVWT), involved segments, maximal wall thickness, and septal morphology. LVWT represents the hallmark of HMC and, while listed as a major risk factor for SCD in both the 2014 ESC and 2020 US guidelines, the latter only considered massive LV hypertrophy ≥ 30 mm in any LV segment as a major risk factor for SCD. In the HCM Risk-SCD model proposed by ESC 2014 guidelines, maximal LVWT was considered as a continuous rather than a dichotomous variable [12,13]. Different hypertrophy patterns have been observed. Based on the location of LV hypertrophy, Maron et al. initially proposed a four-type classification: Type I hypertrophy involves the basal septum; type II involves the whole septum; type III involves the anterior and anterolateral walls of the septum; and type IV involves LV apex. A five-phenotype classification has recently been suggested. It includes the following: type A, predominant mid-septal convexity toward the LV cavity (reverse septum HCM); type B, septum concavity toward the LV cavity and a prominent basal septal bulge (sigmoid septum HCM); type C, an overall straight septum (neutral septum HCM); type D, predominant apical distribution of hypertrophy (apical HCM); and type E, predominant hypertrophy at the mid-ventricular level (mid-ventricular HCM). Interventricular septum (IVS) morphology has been correlated with the probability of a positive genetic test for sarcomeric mutations: accordingly, a reverse IVS is associated with a high probability of a positive genetic test, apical or neutral IVS with a moderate probability, and a “sigmoid” IVS with a low probability of a positive test [18]. The addition of contrast echocardiographic agents could be useful in order to diagnose other localized forms of HCM (such as apical or inferolateral).(2)LV apical aneurysm: US 2020 guidelines consider the presence of an LV apical aneurysm independent of its size as a major risk factor for SCD since it offers the substrate for re-entrant VT. Indeed, the prognosis of HCM patients with LV apical aneurysms is generally unfavorable, with an overall rate of life-threatening complications between 6% and 10% per year, mostly consisting of arrhythmic SCD and thromboembolic events [19,20].(3)The mitral valve, its apparatus, and left ventricular outflow tract obstruction. More than 50% of HCM patients have abnormal mitral leaflets, and more than 25% show abnormalities of the chordae and papillary muscles as a primary phenotypic expression of HCM that may have a pivotal role in left ventricular outflow tract (LVOT) obstruction [21]. One of the parameters of the HCM Risk-SCD model is the maximal LV outflow tract (LVOT) gradient, while US guidelines, although not considering LVOT gradient in SCD risk, include the exercise-induced hypotension that represents its hemodynamic consequence. LVOTO at rest is present in about one-third of HCM patients and is an independent determinant of adverse prognosis [22]. In another one-third of HCM patients, LVOTO is only seen after provocative maneuvers, and treadmill echocardiography (EE) is a key method in detecting an inducible obstruction in HCM [13].(4)LV systolic function: US 2020 guidelines consider decreased LV systolic function (ejection fraction < 50%) as a major risk factor for SCD in HCM patients. However, the limitations of LVEF are well known when LVH is present. EF, mostly reflecting radial wall thickening, is often preserved in HCM, compensating for the reduced longitudinal function seen in this disease. Doppler myocardial imaging (DMI) and 2D speckle-tracking echocardiography (2D-STE) overcome some of these pitfalls [23]. As early signs of LV systolic dysfunction, HCM patients show a decrease in regional and global longitudinal strain (LS) before the impairment of LVEF. A decreased septal and regional LS (>10%) has been related to susceptibility to ventricular arrhythmias in HCM [24]. LV-GLS is significantly related to an increased risk of SCD events [25] and is an independent predictor of appropriate ICD therapy [26]. A 3D echo provides potential further insights on LV function in HCM, showing good correlations with CMR. Three-dimensional strain echocardiography also represents a promising tool, although it is not yet defined as useful in HCM risk stratification [27].(5)LV diastolic function and left atrial volume: HCM is classically defined as a “diastolic disease” and the hallmark of diastolic HF [28]. Nevertheless, no single non-invasive echo Doppler parameter has been validated to be completely accurate in the assessment of LV filling pressures (LV-FPs) in HCM. Although the ESC risk score does not consider diastolic function, left atrial (LA) diameter, which represents one of its main structural consequences, is included. On the other hand, in the US SCD risk evaluation, diastolic dysfunction was not considered. The 2D echo LA volume indexed to body surface area (LAVI, mL/m2, in the four-chamber view) is a simple and mandatory parameter for assessing diastolic function in HCM patients. Moreover, strain analysis of the LA may represent a novel promising tool in HCM risk stratification; in fact, atrial LS was correlated with HF symptoms [29].

#### 2.2.2. Arrhythmogenic Cardiomyopathy

According to the 2020 international criteria, the diagnosis of ACM is focused on a multiparametric approach and needs at least one morpho-functional or structural criterion plus a pathogenic mutation [30]. Among these proposed diagnostic criteria, two are the main findings detectable by echocardiography: firstly, the presence of global LV systolic dysfunction, represented by depression of LV ejection fraction or reduction in LV global longitudinal strain, with or without LV dilatation; secondly, the detection of regional LV hypokinesia or akinesia of the free wall, septum, or both, with a preserved LV global systolic performance. Because of their low disease-specificity, they are considered minor morpho-functional criteria for diagnosing “biventricular” or “dominant-left” disease variants [31].

Of note, in patients with laminopathy (LMNA-positive genetic testing), either mild ventricular dysfunction or a non-significant LV impairment could manifest an intense arrhythmic burden [32,33,34]. In these cases, strain echocardiography has the ability to detect subtle changes highlighting regional LV involvement and is useful in early stratification. Moreover, speckle-tracking echocardiography adds information on the extension of the myocardial damage and thus of the arrhythmic risk by the grading of mechanical dispersion; for this reason, the evidence of an abnormal global longitudinal strain (GLS) or higher mechanical dispersion is an index of a larger myocardial injury, thus promoting fibrosis and potential re-entry circuits [35].

#### 2.2.3. Dilated Cardiomyopathy

Beyond EF, several echocardiographic findings have been proposed as risk markers of ventricular tachycardias in patients affected by DCM. Left ventricular strain is known to be a valuable tool for the identification of subtle systolic dysfunction before an overt drop in LVEF; given its higher sensitivity, it has been considered a better marker of SCD risk and of all-cause mortality risk, compared to LVEF and independently of the extent of LGE, in DCM patients [36].

In fact, GLS has shown to be a better SCD predictor in patients with relatively preserved EF; moreover, reversed apical rotation and loss of LV torsion have been associated with more impaired LV function, due to significant LV remodeling, thereby indicating a more advanced disease stage [37].

The echocardiographic examination must be completed with the assessment of the presence of potentially associated left atrium enlargement, dilatation and contractile RV dysfunction, and secondary pulmonary hypertension, which are usually linked to functional mitral regurgitation that DCM patients may develop due to apical tethering of mitral valve leaflets, annular dilatation, or LV dyssynchrony. All these echocardiographic parameters are associated with an increased risk of cardiac death, hospitalization for heart failure, and ventricular arrhythmias [38]. Indeed, stress echocardiography may reveal helpful information, by assessing the presence of contractile reserve, predicting left ventricular reverse remodeling and functional recovery in DCM patients [39].

### 2.3. Step 3: ECG Monitoring and Exercise Testing in SCD Risk Evaluation

ECG monitoring represents an essential step in SCD risk evaluation of patients with defined or suspected CMP since it allows an estimation of the arrhythmic risk with a relatively greater impact than many other tests or evaluations. Premature ventricular complex (PVc) burden, Lown Class, and non-sustained ventricular tachycardia (NSVT) represent the most important markers of SCD risk, with the latter considered in SCD risk evaluation of all CMPs [10]. In fact, NSVT represents a marker that strongly influences the risk of SCD in HCM, DCM, and even ACM, although conflicting data on the independent predictive value of NSVT in ACM have been produced [10,40]. Similarly, PVc burden is not considered as an independent SCD risk factor for both DMC and HCM, although its utility in ACM has been suggested, together with other parameters [41]. Monitoring over a period of 24–48 h is appropriate for daily arrhythmias while monitoring over a longer period should be preferred for infrequent events. Moreover, patient-activated ECG recorders (e.g., mobile-health) may represent a new promising tool in arrhythmic risk evaluation [42]. On the other hand, implantable loop recorders (ILRs) may be useful in diagnosing arrhythmias that have escaped from conventional ECG monitoring, especially in patients with unexplained syncope suspected of arrhythmic cause [43].

Finally, exercise testing provides useful and unique information regarding mechanisms of functional limitation, blood pressure and heart rate response to exercise, and ischemic and arrhythmic burden in CMPs. Moreover, exercise testing is crucial to study the susceptibility to catecholamine-induced arrhythmias in patients with CMPs. In particular, the presence of NSVT during exercise represents a risk marker with a strong prognostic impact on the estimation of arrhythmic risk [44]. In HCM, documented NSVT during exercise testing is very rare, but was associated with a higher risk of SCD [45].

Beyond traditional ergometric tests, cardiopulmonary exercise testing (CPET) represents a valuable safety tool in all CMPs for assessment of functional capacity (NYHA Class quantification) and exertional arrhythmic risk as well as for differential diagnosis of angina and dyspnea. In particular, CPET allows differentiating between cardiac and non-cardiac causes of dyspnea as well as evaluating specific mechanisms of breathing discomfort in cardiac patients such as systolic or diastolic dysfunction, mitral regurgitation, chronotropic incompetence, or, as specifically in the case of obstructive HCM, low cardiac output due to significant inducible obstruction. Figure 4 shows a case of an HCM patient with severe LVH, NSVT, and myocardial fibrosis on CMR (right side) without significant LVOT obstruction at baseline that became significant during exercise (left side). More specifically, during exercise, stroke volume (blue line) begins to decline before the lactate threshold (LT) as a consequence of a progressive increase in the LVOT gradient of obstruction. Cardiac output is progressively sustained by heart rate only (red line) as a compensatory response; after the respiratory compensation point (RC), exercise was limited by dyspnea and hypotension associated with a gradient of obstruction of 74 mmHg (documented by echocardiography during exercise).

Finally, low VO2max represents a major predictive marker of adverse outcomes in all CMPs as well as in heart failure patients [46].

### 2.4. Step 4: Multimodality Imaging in SCD Risk Evaluation

A stepwise multimodal approach is recommended, beginning with a 2D echocardiogram evaluation and possibly implementing additional echocardiographic methods such as TDI, strain, automatic ejection fraction (EF), 3D echocardiogram, and contrast echocardiogram. Following that, second-level imaging tests such as cardiac magnetic resonance (CMR), cardiac computed tomography (CCT), and cardiac nuclear imaging could be used to better define the anatomical and functional characteristics of CMPs and thus further stratify the risk of SCD and the patient’s prognosis.

CMR should be considered in all patients and should be performed at least once (at the initial evaluation) and may be repeated according to potential changes in the clinical status, in order to answer specific clinical questions and problems. Cardiac CT and nuclear imaging techniques have more limited indications and are only indicated in specific clinical situations such as angina or angina equivalents.

#### 2.4.1. Hypertrophic Cardiomyopathy

In adult patients with HCM, diffuse fibrosis is a predictor of non-sustained VT and aborted SCD [47]. Moreover, a linear relation has been demonstrated between the amount of LGE (in terms of percentage of myocardial mass) and SCD risk [48]. Interestingly, CMR feature tracking (CMR-FT)-derived GLS is also a strong independent predictor of major adverse cardiac events, including hospitalization for heart failure, resuscitated cardiac arrest due to ventricular tachyarrhythmias, and SCD [49]. New promising CMR methods include T1 mapping; a prolonged myocardial T1 with elevated extracellular volume in patients with HCM suggests diffuse myocardial fibrosis, even in the absence of regionally definite LGE and hemodynamic LVOT obstruction [50]. In fact, a computational modeling approach that merges data from LGE-CMR with data from post-contrast T1 mapping is able to reveal extensive diffuse fibrotic remodeling, which is a risk factor for SCD and ventricular tachyarrhythmia in HCM and is associated with the occurrence of new ventricular tachyarrhythmias [51]. Moreover, a recent study found in a large cohort of nonischemic cardiomyopathy patients that extracellular volume (ECV) was the only parameter that demonstrated an independent and strong predictive value for ventricular arrhythmias and SCD, on top of LGE and LVEF. The best cut-off of ECV to predict the arrhythmic outcome was 30%. ECV ≥ 30% improved the risk stratification among LGE+ patients and among those with LVEF ≤ 35% [52]. Recently, the measurement of entropy was applied to evaluate heterogeneity in fibrotic lesions [53]. Scar heterogeneity, which is quantified by determining the entropy within a scar, and LGE extent were shown to be independent risk indicators of ventricular arrhythmias [54].

Imaging tests aimed at the study of myocardial blood flow and coronary flow reserve are useful in the evaluation of patients with HCM since myocardial ischemia represents a common feature in all stages of the disease and confers prognostic information [55]. In HCM, both severe structural remodeling of intramural coronary arterioles and increased perivascular fibrosis have been described; these structural changes, in addition to the increased extravascular compression, represent the basis of coronary microvascular dysfunction in HCM [17]. The degree of hyperemic myocardial blood flow impairment is a powerful long-term predictor of adverse LV remodeling and systolic dysfunction, as well as an independent predictor of death and unfavorable outcomes in HCM [56].

#### 2.4.2. Arrhythmogenic Cardiomyopathy

Tissue characterization by CMR represents the imaging key diagnostic tool for disease confirmation and for SCD risk stratification in terms of the extent of LGE in ACM.

The 2020 international criteria conferred a prominent role in the identification of LV “stria” or “ring-like” LGE pattern in one or more myocardial segments of the free wall, at the subepicardial or mid-myocardial level. The above-mentioned LGE patterns are considered the only major diagnostic criterion for the diagnosis of ACM, with the exclusion of a “junctional” LGE pattern, characterized by focal/patchy involvement of the posterior or less frequently anterior ventricular septum at the site of RV attachment [31].

Tissue characterization plays a key role in the identification of LGE/fibrosis which is the most sensitive feature for the diagnosis of ACM and useful for SCD risk stratification; a diffuse LGE has been associated with an increased risk of life-threatening ventricular arrhythmias and has been considered in the 2020 ESC guidelines’ algorithm as a marker of SCD in patients with LVEF between 35% and 50%.

In recent years, LV myocardial strain and dyssynchrony detected by CMR feature tracking (CMR-FT) have been defined as additional diagnostic tools in predicting adverse cardiovascular outcomes and sudden cardiac death risk in ACM [57].

Regarding other imaging techniques, the role of cardiac nuclear imaging in the diagnosis and risk stratification of patients affected by ACM is still limited. Only in carriers of desmoplakin (DSP) gene variants, characterized by predominant LV involvement and by recurrent episodes of acute chest pain associated with myocardial injury and/or ventricular arrhythmias, 18-FDG PET may improve the assessment of inflammatory “warm phase”, which represents the early pre-phenotypic stage of ACM [58]. According to a recent study, patients with DSP-related cardiomyopathy usually demonstrate an 18-FDG PET-positive scan in 59% of cases, in terms of focal (70%) or diffuse (30%) uptake; in this sense, it has been suggested to apply FDG-PET as a method to differentiate diagnosis in the context of ACM phenocopies, such as myocarditis, cardiac sarcoidosis, and other immune-mediated diseases [59,60,61].

#### 2.4.3. Dilated Cardiomyopathy

LGE is present in around 30% of patients with DCM, typically in a mid-wall pattern. It has been associated with the occurrence of major arrhythmic events consistently across all subgroups of DCM, regardless of LVEF. Idiopathic DCM is typically associated with septal mid-wall LGE; this specific pattern of distribution, with or without free-wall involvement, has been associated with the highest mortality SCD risk [62]. More specifically, the presence of concomitant septal and free-wall LGE or of multiple LGE patterns (mid-wall laminae, subepicardial striae, or subendocardial focal enhancement patches) has been linked to a higher risk and an increased risk of all-cause mortality [63]. In fact, myocardial fibrosis represents the organic substrate around which an arrhythmic circuit can perpetuate itself as well as a histopathological aspect associated with progression towards dysfunction, heart failure, and ultimately cardiovascular mortality.

Leading to poor prognosis, an overlapping DCM/ACM phenotype can be recognized in the presence of a subepicardial, ring-like scar pattern, associated with a more focal LV impairment and correlated with specific mutations, such as DSP and FLNC genotypes [64]. Moreover, higher native-T1 myocardium values constitute a notable predictor of all-cause mortality, arrhythmic outcomes, and HF risk in patients with DCM [65]. Finally, feature-tracking (FT) strain analysis could be a promising technique for improving risk stratification of DCM patients undergoing CMR; in fact, independently of LVEF, LGE, and clinical parameters, CMR-FT findings are associated with a better cardiovascular outcome in the presence of preserved FT-GLS, even in patients with reduced LVEF < 35% and/or with evidence of LGE [66].

Considering other imaging techniques, nuclear imaging (e.g., PET and SPECT) may provide useful information in sudden death risk stratification, although nowadays its prognostic impact and applicability are more limited. In particular, since abnormal cardiac sympathetic tone represents one of the most relevant factors potentially inducing ventricular arrhythmias, SPECT with 123I-MIBG seems to provide additive prognostic impact in DCM allowing the assessment of sympathetic innervation at the cardiac level, which has been shown to be associated with arrhythmic events and disease progression [67].

### 2.5. Step 5: Genetic Testing in SCD Risk Evaluation

In the field of cardiomyopathies, genetic evaluation is of crucial importance for both diagnostic and prognostic purposes [16]. Nevertheless, the availability of genetic tests as well as their costs represent a barrier to their widespread use. Furthermore, their interpretation profoundly depends on the accuracy of the clinical and phenotypic evaluation of cardiomyopathy performed before genetic testing. For this reason, although of crucial importance, the genetic test is proposed in the present hierarchical approach at the end of the clinical and instrumental evaluation.

Although until a few years ago the idea of implanting a defibrillator based on a genetic test would have been considered “science fiction”, nowadays genetic testing is showing a progressively greater impact, especially in the case of mutations with a very high impact on survival free from SCD, such as LMN, Filamin C (*FLNC*), and Phospholamban (PLN gene) [10]. Wide application of genetic testing nowadays frequently allows the identification of specific genetic single-nucleotide polymorphisms causative for different CMP phenotypes.

Moreover, polygenic risk scores may be useful not only in CMP diagnosis but also in CMP prognostication [68]. A standardized approach to the interpretation of the pathogenicity of genetic variants has been widely accepted and allows the differentiation of five classes of mutations:Class V: pathogenic;Class IV: likely pathogenic;Class III: variant of uncertain significance;Class II: likely benign;Class I: benign [69].

A class IV or V mutation allows the confirmation of a diagnosis in probands (the first affected family member) and could be crucially important for the early diagnosis of relatives. Since the pathogenicity of each specific mutation is subject to continuous updating, a periodic reassessment of all mutations, especially for class IV and III variants, should be performed [69]. A negative result does not exclude a diagnosis and should not be used for this purpose. Since autosomal dominant inheritance represents the most frequent inheritance pattern of all CMPs, cardiac screening in first-degree relatives is always recommended [69].

In HCM, sarcomere gene mutation can be identified in 30–60% of patients. The most common HCM mutations are as follows: β-Myosin heavy chain (MYH7), Myosin binding protein-C (MYBPC3), Cardiac troponin T (TNNT2), Cardiac troponin I (TNNI3), Cardiac α-actin (ACTC1), α-tropomyosin (TPM1). Those associated with high SCD risk are mainly MYBPC3 and MYH7 [70].

Conversely to HCM, both genetic and structural overlap are not infrequent in DCM and ACM [71].

In particular, in DCM and ACM, a wide variability in phenotypic expression can be found within the same family or the same patient over time. Next-generation sequencing allows the identification of a causative monogenic variant in approximately 30% to 45% of cases of DCM and ACM [72]. In particular, DCM/ACM with LMNA mutations, which represent 5–10% of all DCM patients, are associated with early atrial and ventricular arrhythmias, premature conduction disease, a high risk of SCD, and progression to end-stage heart failure [73,74].

In conclusion, the impact of genetics not only on the diagnosis but also on the prognosis of patients with CMPs is increasingly greater. The presence of a positive genetic test represents an additional risk marker beyond the traditional clinical and instrumental markers. The presence of multiple unidentified mutations in the same proband is associated with increased risk.

## 3. The World of ACM and DCM: How “Complex Imaging” (Electroanatomic Mapping) Directs Diagnosis and Invasive Management

This section concerns mapping and ablation in cardiomyopathies.

### 3.1. Arrhythmogenic Cardiomyopathy

Despite the presence of standardized diagnostic criteria, the diagnosis of ACM remains a clinical challenge, specifically at its early stage, or in the “formes frustes” [75].

Routine imaging techniques, and in particular CMR, may lead to misdiagnosis of ACM by showing equivocal morpho-functional RV abnormalities: fat and fibrosis give the same signal on CE-CMR, and partial volume effects make it difficult to distinguish between the two different tissues, mostly in a thinned wall.

In sudden cardiac death risk evaluation, three-dimensional electroanatomic voltage mapping (EVM) offers the potential to identify the presence, location, and extent of the pathological substrate of ACM by detection of low-voltage areas that correspond to regions of RV myocardial loss and fibrofatty replacement [76]. Corrado et al. tested the hypothesis that RV electroanatomic low-amplitude areas were significantly associated with the histopathological finding of myocyte loss and fibrofatty replacement in endomyocardial biopsy [76]. This peculiar pathological process leads to islets of residual myocytes interspersed among adipocytes and fibrous tissue, providing an ideal milieu for re-entrant VT [77]. Similarly, case series suggested that low-voltage areas on invasive EVM could identify ACM at an earlier stage as compared to LGE and fatty infiltration on CE-CMR [78].

This “electroanatomic scar” area is defined as an area ≥ 1 cm^2^, including at least three adjacent points with bipolar signal amplitude < 0.5 mV. The color display for depicting normal and abnormal voltage myocardium ranges from “red” representing “electroanatomic scar tissue” (amplitude < 0.5 mV) to “purple” representing “electroanatomic normal tissue” (amplitude ≥ 1.5 mV). Intermediate colors represent the “electroanatomic border zone” (signal amplitudes between 0.5 and 1.5 mV) (1,3). A relatively sharp border, as identified by a steep spatial voltage gradient, could be used to demarcate the dysplastic regions. Unipolar voltage (UV) mapping of RV endocardium has been explored to predict the disease and the epicardial arrhythmogenic substrates. Evaluation of the epicardial abnormal substrates can be achieved using the RV endocardial unipolar voltage mapping with a cut-off value of 5.5 mV, and the abnormal area is correlated to the epicardial scar in ACM, although the different cut-off value of 4.4 mV has been proposed through the site-by-site comparison [79]. The previous studies based on CE-CMR imaging also supported that the bipolar low-voltage area and the territories displaying abnormal EGMs were correlated to the transmural scar [79,80]. In addition, even in the absence of an RV scar in CMR, parameters significantly associated with electroanatomic low-voltage areas are right precordial QRS prolongation, low-voltage QRS in the limb leads, and late potentials. Accordingly, intracardiac EGMs recorded from within the electroanatomic RV low-voltage areas often appeared fractionated with significantly prolonged duration and extended beyond the offset of the surface QRS compared with EGMs recorded from normal-voltage areas [78]. Hence, the ability to identify the dysplastic process by the presence of low-amplitude EGMs may be used as a new criterion in these patients’ work-up [75]. Considering the extensive pathological substrate due to the fact that the disease process in ACM initiates from the epicardium toward the endocardium, it is important to emphasize the relevance of a comprehensive and extensive substrate-based ablation strategy that incorporates endocardial (ENDO) and, if still inducible, epicardial (EPI) ablation to achieve the long-term VT control [77]. The 2022 ESC guidelines for the management of patients with ventricular arrhythmias and the prevention of sudden cardiac death recommended (class IIa, level of evidence C) ENDO and adjuvant EPI substrate ablation in patients with ACM and recurrent, symptomatic sustained monomorphic VT (SMVT), ICD shocks for SMVT despite anti-arrhythmogenic drugs or beta-blockers, or persistent inducibility after ENDO-only ablation.

The outcome of VT ablation in ACM has been investigated in multiple previous studies. The reported freedom from VT following ablation in these studies is between 45% and 85%, with variable procedure methods and follow-up periods [81].

### 3.2. Dilated Cardiomyopathy

In sudden cardiac death risk evaluation of both DCM and nonischemic dilated cardiomyopathy (NIDCM), EVM is of crucial importance in order to assess the presence of underlying arrhythmic substrates, which are typically located at the basal perivalvular regions and at the interventricular septum, with a high prevalence of mid-myocardial and subepicardial substrates [82].

The first mapping studies in patients with DCM demonstrated that >80% of SMVTs are due to myocardial re-entry [83,84] and occasionally to triggered activity, both of which are associated with the presence of scarring [85]. The re-entry circuits are usually associated with regions of low-amplitude EGMs, consistent with the scar, in agreement with the findings of Hsia et al. Early studies by Cassidy et al. [86], demonstrated fewer abnormal EGMs in patients who had cardiomyopathy compared with patients who had a myocardial infarction. The cellular mechanisms responsible for differences in scar formation are unknown. As the scar in NIDCM tends to be smaller and less confluent than in ischemic cardiomyopathy (ICM), the re-entrant circuit may have different anatomic and functional properties that affect propagation [87]. Thus, patients with DCM result to have less dense electroanatomic scars: late potentials cannot be identified in >50% of patients with DCM. The lack of abnormal EGMs has been attributed to deep intramural substrate location [85].

As in ACM, UV mapping has been reported to have a larger field of view and might be able to detect intramural and subepicardial scar patients with DCM. Normal values are based on mapping studies in healthy control subjects, with 95% of all voltages being >8.27 mV [88]. Areas without enhancement from LGE-CMR integrated with EVM have also been used as a reference for normal myocardium [85]. Studies found two distinct patterns of myocardial scarring in patients with DCM, namely anteroseptal and inferolateral [89], and the regions of scarring were frequently adjacent to a valve annulus, as is often the case in VT after inferior wall infarction [84]. The annulus often seems to form a border for an isthmus in the re-entry path. It is interesting to speculate that the formation of a long channel, or isthmus, along an annulus contributes to the formation of re-entry circuits that can support sustained monomorphic VT [83]. The precise cause of distinct scar distributions in NICM is unclear, possibly relating to a combination of excess basal mechanical stress and underlying cardiomyopathic processes [89]. The scar distribution and the ablation outcome are also different depending on the underlying etiology. EVM performed in 25 patients with LMNA mutation DCM confirmed dominant basal septum involvement in 78% of patients, followed by the subaortic LV (53%), the mid-septum (50%), and the basal inferior wall (39%). Radiofrequency catheter ablation (RFCA) outcome in LMNA cardiomyopathy is poor. Procedural success after multiple procedures, including bailout strategies (trans-coronary ethanol ablation in 25%, surgical ablation in 8%), could be achieved in only 25% [90].

Current guidelines [10] recommend RFCA as adjunctive treatment to prevent recurrent ICD therapies for SMVT that cannot be controlled by amiodarone or sotalol independent of the underlying etiology. Evidence supporting the benefit of RFCA mainly stems from small retrospective studies, with discordant outcomes and arrhythmia-free survivals ranging from 30% to 71% [77,86,89,91]. A more recent study that involved 282 patients confirms that endocardial with adjuvant epicardial mapping and ablation when indicated (i.e., early recurrence of VT or persistent inducibility after endocardial-only ablation) provides good long-term outcomes [82].

However, in consideration of the scar extension and distribution in DCM, Soejima et al. found that the success of endocardial ablation was lower than that they observed for post-infarct VT. Re-entry circuits deep to the endocardium and in the epicardium appear to be a likely explanation. Epicardial mapping led to successful ablation in more than half of the patients in whom it was attempted. Despite frequent VT terminations during ablation, recurrences are frequent and not reliably predicted by post-ablation programmed stimulation. The intramural scarring process typical of DCM does not allow the identification of VT isthmuses in many instances and also prevents effective and long-lasting lesion formation because of the limited (<5 mm) penetration of the radiofrequency (RF) energy. Although alternative energy delivery modalities have been described, including bipolar RF, high-intensity focused ultrasound, needle ablation, and intracoronary ethanol injection, none of them are currently available for routine clinical application, and they should be considered as bailout strategies in patients who need epicardial ablation with pathogenic mutation (LMNA gene) and in those with deep intramural, anteroseptal substrate location [89] (Figure 5).

### 3.3. ICD Implant in Cardiomyopathies

Ventricular arrhythmias (VAs) manifesting as SCD are one of the main causes of death in patients with cardiomyopathies. Recent guidelines, for the first time, extend the flowchart for SCD/VA risk assessment and ICD implantation to a strongly multiparametric approach with an increased role of complex imaging and genetic evaluation.

Historically, traditional risk markers of SCD in HCM include recent unexplained (non-vasovagal) syncope, family history of juvenile SCD, episodes of non-sustained ventricular tachycardia (NSVT), massive left ventricular (LV) hypertrophy, and exercise-induced hypotension [13]. The 2020 US guidelines still endorse a single risk factor-based decision model that has been improved with the addition of recently introduced markers of SCD: extensive late gadolinium-enhancement on cardiac magnetic resonance (>15%), advanced heart failure (with LVEF < 50%), and left ventricular (LV) apical aneurysm [12]. Conversely, in 2014, the European Society of Cardiology (ESC) guidelines abandoned risk assessment strategies based on counting or weighting single risk factors and endorsed a 5-year SCD risk stratification score based on seven factors (age, LV wall thickness, LA size, LVOT gradient, NSVT, unexplained syncope, and family history of SCD) (HCM Risk-SCD: https://doc2do.com/hcm/webHCM.html, accessed on 5 April 2023) [11,92]. This calculator is not intended for use in elite athletes, in individuals with metabolic or infiltrative diseases, after myectomy, or after alcohol septal ablation. According to current guidelines, in patients aged 16 years or more, ICD implantation should be considered with an estimated 5-year risk of SD ≥ 6% based on the HCM Risk-SCD (class of recommendation IIa, level of evidence B). Implantation should also be considered in patients with an intermediate (≥4 and 6%) 5-year risk of SCD with other risk factors for SCD such as significant LGE in CMR (usually ≥15% of LV mass), LVEF < 50%, abnormal blood pressure response during exercise test, LV apical aneurysm, or presence of sarcomeric pathogenic mutation (class of recommendation IIa, level of evidence B). ICD may also be considered, but with a lower level of recommendation, with an estimated 5-year risk of SCD ≥ 4 to <6% (class of recommendation IIb, level of evidence B). Hence, implantation may be considered with a low estimated 5-year risk of SCD (<4%) associated with another risk factor such as significant LGE in CMR (usually ≥15% of LV mass), LVEF < 50%, or LV apical aneurysm (class of recommendation IIb, level of evidence B) [10].

Figure 6 illustrates, considering HCM as an example, the various factors to take into consideration while making the challenging choice of whether to implant an ICD. In particular, the choice of ICD implant depends on an accurate evaluation of the balance between benefit and risk for each subject, thereby considering the relative impact of SCD risk factors as well as the procedural or long-term risk of ICD complications. Individual HCM risk factors for SCD are ranked according to their prognostic impact; moreover, the three risk classes according to HCM risk score are also displayed (upper side) [13]. Additionally, the relative sensitivities and specificities of the two models provided by the ESC and the US are shown together with the annual rate of appropriate or inappropriate ICD interventions and ICD complications (lower side) [93].

Finally, the role of genotyping for risk stratification in HCM remains uncertain. Patients with pathogenic sarcomere gene mutations seem to have an earlier and more severe phenotypic expression and a worse outcome with increased overall mortality, increased risk of ventricular arrhythmias and SCD, and greater incidence of atrial fibrillation and heart failure than sarcomere-mutation-negative HCM patients. However, the prognostic role of genetic background is limited by the high variability of the clinical impact of the same genetic variant, either within or between families. So, actually, results of genotyping are not generally used for risk stratification of SCD and therapy decision making, with particular reference to ICD implantation [10].

In DCM, according to current guidelines [10], ICD implantation should be considered in patients symptomatic for heart failure with NYHA class II–III, and a reduction in LVEF ≤ 35% after almost three months of OMT (class of recommendation IIa, level of evidence A).

For DCM associated with LMNA mutation carriers, a risk calculator has recently been developed (https://lmna-riskvta.fr/, accessed on 5 April 2023) to predict the risk of life-threatening VA [11]. To avoid over-implantation in mutation carriers without a cardiac phenotype, according to current guidelines [10], a primary prevention ICD implantation should be considered in patients with a 5-year estimated risk ≥10% and a manifest cardiac phenotype (NSVT, LVEF < 50%, or AV conduction delay) (class of recommendation IIa, level of evidence B). ICD implantation should also be considered in DCM patients with a preserved EF (LVEF ≥ 50%) and two or more of the following risk factors: history of syncope; LGE on CMR; inducible SMVT; and pathogenic mutations in LMNA, PLN, FLNC, and RMB20 genes (class of recommendation IIa, level of evidence C) [10]. Nowadays, the accuracy of arrhythmic SCD risk stratification for primary prevention in this cohort of patients remains suboptimal. In fact, the recent DANISH trial showed no reduction in the primary endpoint of all-cause death in patients with nonischemic systolic heart failure (LVEF ≤ 35%) randomized to ICD implantation or not after optimal medical therapy (OMT), despite a significant reduction in SCD in the ICD group. The DANISH trial also showed that the rate of appropriate ICD shocks was only 11% in a follow-up of 5 years, failing to confirm any additional benefit of ICD therapy in a cohort of nonischemic cardiomyopathy (NICM) patients in which implantation was merely based on LVEF [94].

In ACM, the risk of VA is more consistent than that in any other cardiomyopathy. In ACM patients not implanted with ICDs, cardiac arrest (CA) occurs in 4.6–6.1%, while 23% of patients experience a non-fatal MSVT during an average follow-up of 8–11 years [95]. In a large cohort of ACM patients, 43% of them had VT/VF during a median follow-up of 5.75 years, but only 10.8% had a potentially life-threatening event. A relevant role in ICD indications is played by arrhythmic syncope, which has been shown to be a predictor for subsequent events in most series of patients with definitive ACM [96]; for this reason, in these patients, an ICD should be considered according to current guidelines (class of recommendation IIa, level of evidence B) [10].

In addition, in ACM patients, RV and LV dysfunction have been associated with a higher arrhythmic risk [97]. Cut-off values are difficult to determine, but in patients with severe RV dysfunction (RV fractional area change ≤17% or RV ejection fraction ≤ 35%), an ICD implantation in primary prevention should be considered (class of recommendation IIa, level of evidence C). According to current guidelines, ICD implantation should also be considered in symptomatic ACM patients with moderate RV (RV ejection fraction < 40%) or LV dysfunction (LV ejection fraction < 45%) and who have inducible NSVT or MSVT (class of recommendation IIa, level of evidence C) [10]. Hence, the majority of VA episodes in ACM ICD recipients are MSVT (Up to 97%). The very high termination rate by ATP (92% of all episodes) independent of VT cycle length strongly supports the implantation of devices with the capability for ATP in ACM patients [98]. Lastly, gene-specific risk stratification of ACM is still a matter of debate. Gene positivity for pathogenic mutations seems to confer a worse outcome, particularly for LMNA. In addition, in ACM, specific mutations may correlate with earlier manifestation phenotypes and may influence risk stratification and management [74]. Considering the prognostic profile of gene positivity, the identification of some pathogenic mutations may help decision making about primary ICD implantation [99]. With regard to non-desmosomal gene defects, the TMEM43 p.S358L founder mutation is almost fully penetrant and highly lethal among male carriers and should be to be considered by itself an indication of prophylactic ICD [100].

## 4. Conclusions

The present review, examining each risk factor of SCD in primary CMPs, proposes a hierarchical approach for the global estimation of SCD risk that starts from the clinical evaluation, an essential and highly informative step, and subsequently passes through the role of electrocardiographic monitoring (both 24 h and during exercise) and multimodality imaging, finally concluding with genetic evaluation. In fact, the SCD risk assessment in CMPs depends on a multiparametric approach, thereby overturning the historical “LVEF-centric view” of ischemic heart disease (Figure 7).

Of note, SCD risk estimation requires a multiparametric approach and at the same time requires a continuous re-evaluation since specific therapies as well as the evolution (benign or malignant) of the disease may strongly modify the risk. Similarly, the passage of time, namely age, also represents a parameter of great importance in the evaluation of SCD risk, since a benign form generally manifests itself in older age or does not show any evolution over the years. For these reasons, in SCD risk of CMPs, there is an urgent need for more precise and patient-tailored risk stratification.

The present review also highlights the growing importance of genetics and the genotype–phenotype relationship in assessing the risk of SCD in CMPs. Finally, the role of interventional treatment may positively impact SCD in CMPs, as in the case of alcohol septal ablation or surgical myectomy to treat LV obstruction in HCM or in the case of an effective VT ablation in all CMPs as well as a prompt and appropriate ICD implant.

In conclusion, although cardiomyopathies may present several different phenotypes as well as many different genotypes often overlapping each other, an overlap of SCD risk factors is also present among all the different forms of CMPs. This overlap of risk factors is mainly represented by the following great/universal stratifiers: age at presentation, family history of SCD, unexplained syncope, heart failure symptoms, NSVT, ejection fraction, and LGE in CMR.

## Figures and Tables

**Figure 1 jpm-13-00877-f001:**
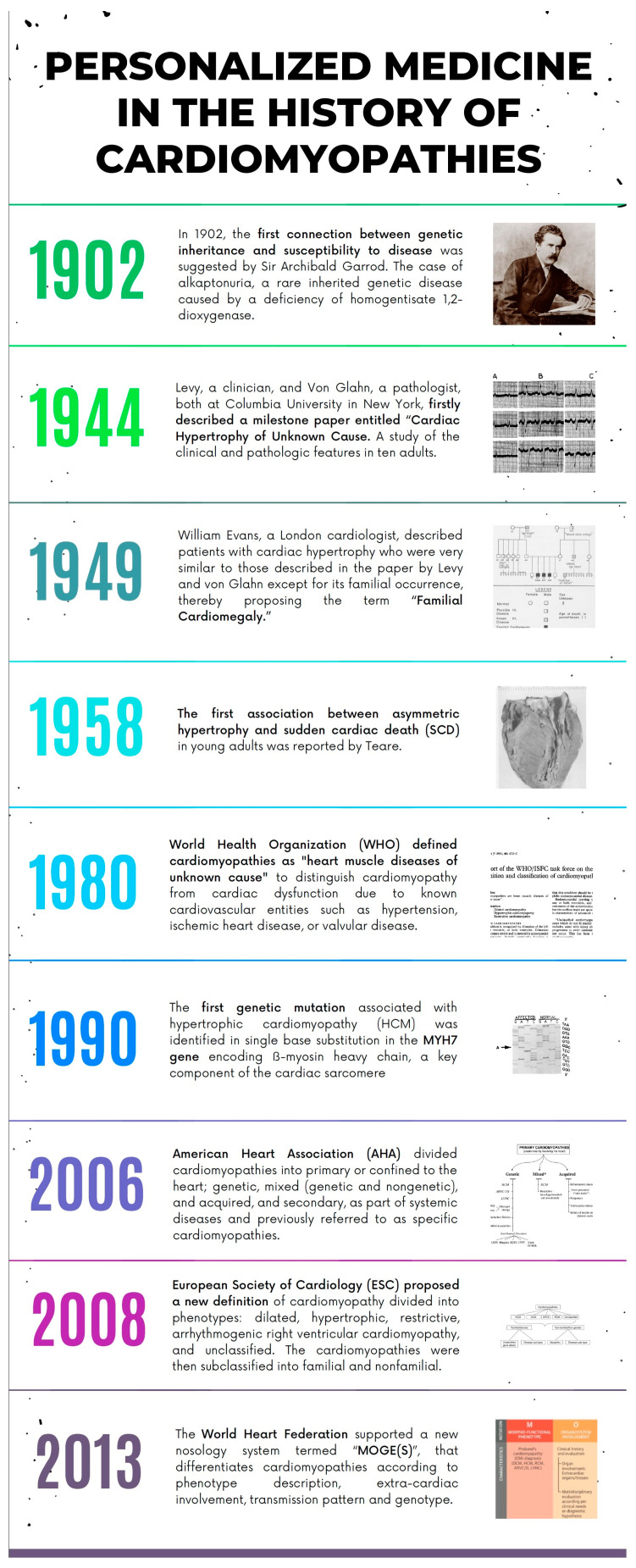
The most relevant moments in the history of personalized medicine in the field of cardiomyopathies.

**Figure 2 jpm-13-00877-f002:**
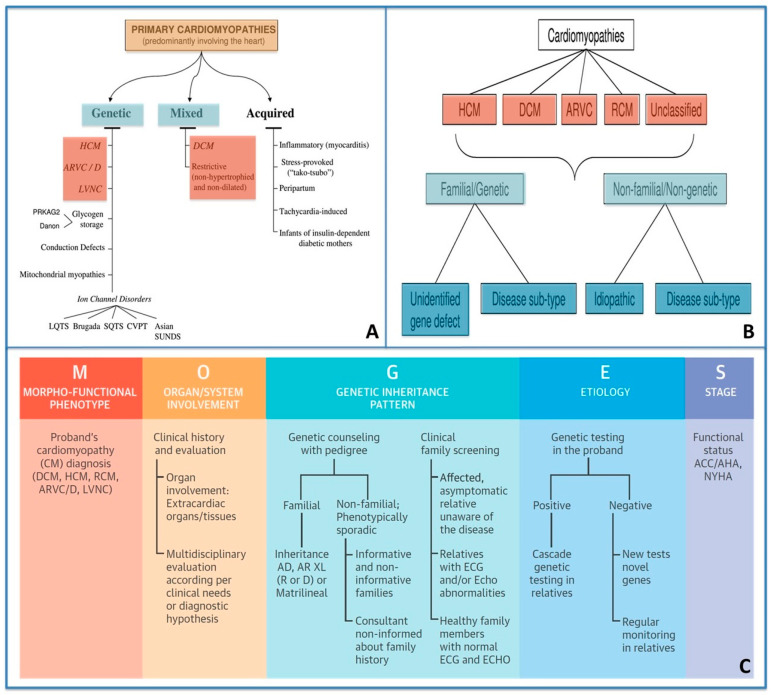
The three current different classifications of cardiomyopathies: US (panel **A**), ESC (panel **B**) and MOGES (panel **C**).

**Figure 3 jpm-13-00877-f003:**
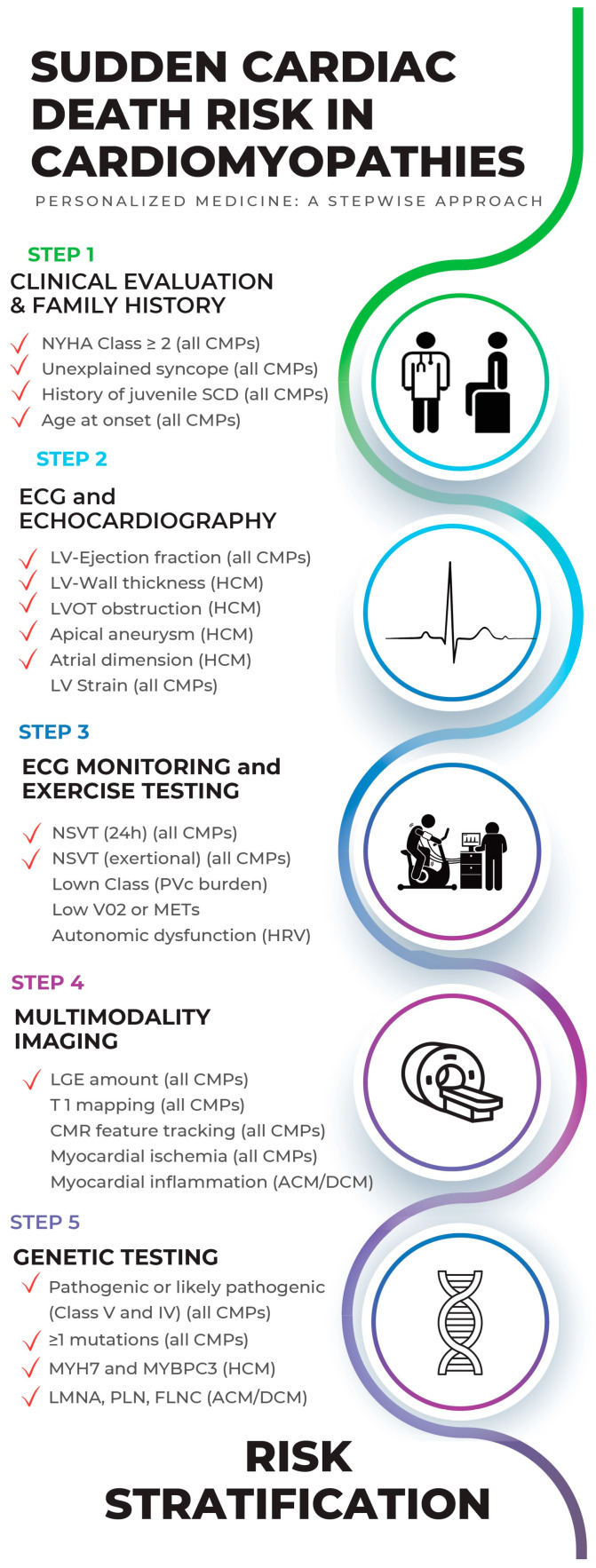
The proposed hierarchical personalized stepwise approach for SCD risk estimation in cardiomyopathies. The red ticks show the risk factors for SCD with the highest evidence and considered in the current guidelines. In brackets, it is indicated if the risk factor applies to all cardiomyopathies or only one of them.

**Figure 4 jpm-13-00877-f004:**
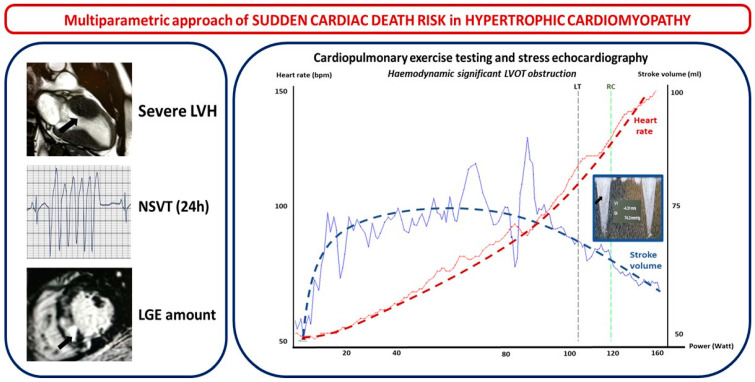
Multiparametric approach of SCD risk in HCM.

**Figure 5 jpm-13-00877-f005:**
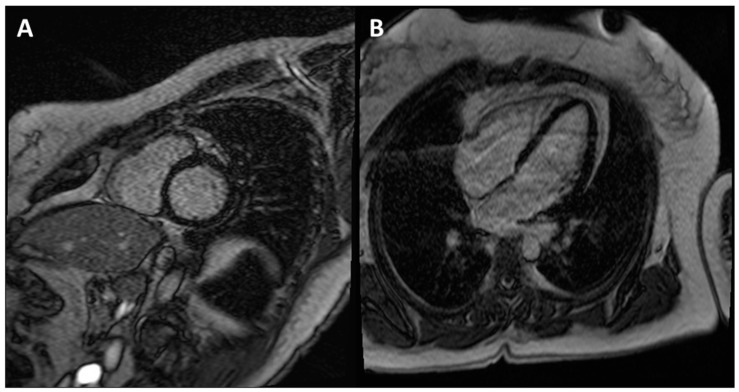
CE-CMR images in patients with ACM without RV morpho-functional abnormalities and LGE. (**A**) Short axis two-chamber view. (**B**) Long axis four-chamber view. (**C**) RV electroanatomic mapping depicting an epicardial scar on the RV free wall. Unipolar voltage map showing low voltages consistent with an epicardial scar. (**D**) Bipolar voltage map showing normal voltages.

**Figure 6 jpm-13-00877-f006:**
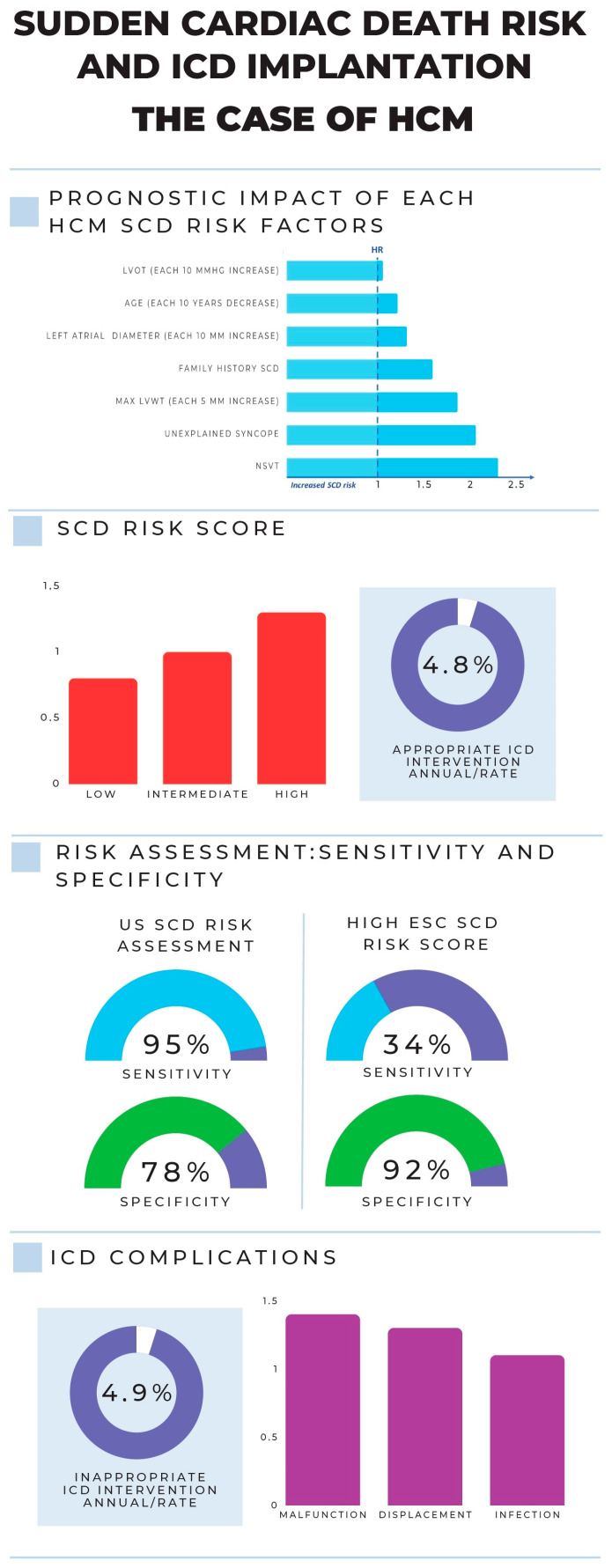
The prognostic impact of each SCD risk factor in HCM, the three ESC SCD risk classes, the percentage/year of appropriate ICD interventions, the different sensitivities and specificities of the ESC and US models, and finally the ICD complication rates.

**Figure 7 jpm-13-00877-f007:**
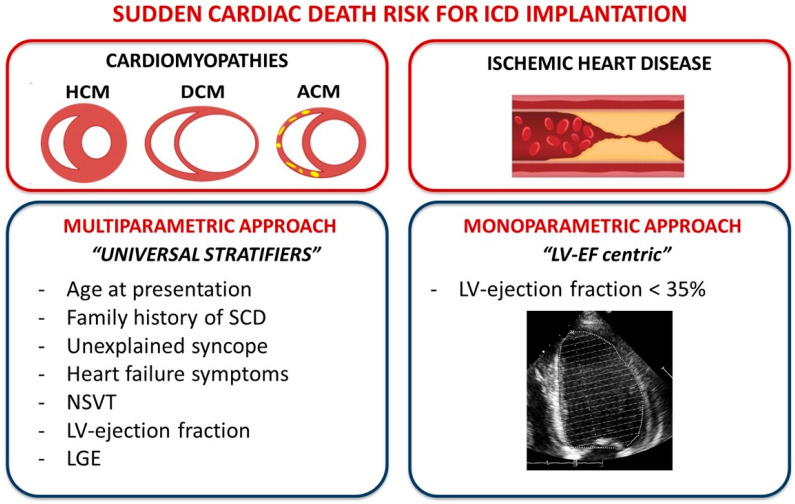
The different approach of SCD risk assessment between ischemic heart disease and CMPs: a multiparametric approach in CMPs vs. the historical “LVEF-centric view” of ischemic heart disease.

**Table 1 jpm-13-00877-t001:** Differential diagnosis between different forms of left ventricular hypertrophy from hypertensive heart to HCM and HCM phenocopies.

	Inheritance	Signs or Symptoms of Multiorgan Involvement	ECG Abnormalities beyond LVH Criteria	Routine Laboratory Tests	Echocardiography	CMR (LGE)
Athlete’s heart	None	Uncommon	Isolated LVH	Not specific	LVH symmetrical or eccentric (mild-to-moderate); Normal systolic and diastolic function	Negative
Hypertensive heart	None	Uncommon	ST and T abnormalities	Not specific	LVH usually concentric (mild-to-moderate)	Mild degree; no specific pattern
HCM	AD	Uncommon	High LVH; ST and T abnormalities; Giant T wave inversion; Q waves	Not specific	Moderate-to-severe LVH (*asymmetrical and septal, potentially found at any location*); diastolic dysfunction, LVOT obstruction, mitral valve abnormalities (*mitral SAM, leaflets and chordal elongation, dysplasia, prolapse, hypermobility*); atrial enlargement; apical aneurysm	Frequent; RV insertion points and intramural; potentially found at any location
Anderson–Fabry disease	X-linked	Visual impairment; sensorineural deafness; paresthesiae and sensory abnormalities; angiokeratoma	Short P-R/preexcitation; AV block	Proteinuria with or without glomerular filtration rate	Concentric LVH; increased atrioventricular valve and RV free wall thickness; global hypokinesia (with/without LV dilatation)	Frequent; posterolateral in concentric LVH
Familial amyloidosis	AD	Visual impairment; paresthesiae and sensory abnormalities; carpal tunnel syndrome (bilateral)	Low QRS voltage; AV block	Proteinuria with or without glomerular filtration rate	Increased interatrial septum, atrioventricular valve, and RV free wall thickness; pericardial effusion; myocardium’s ground-glass appearance; global hypokinesia (with/without LV dilatation)	Frequent; diffuse subendocardial “zebra” pattern; intense myocardial “avidity” for gadolinium
Danon disease	X-linked	Learning difficulties, mental retardation; visual impairment	Short P-R/preexcitation; AV block; extreme LVH (Sokolow > 100)	↑Creatine kinase ↑Transaminase	Extreme concentric LVH; global hypokinesia (with/without LV dilatation)	Frequent; large amount subendocardial or transmural
Mitochondrial CMP	X-linked or matrilinear	Sensorineural deafness; learning difficulties, mental retardation; visual impairment; muscle weakness	Short P-R/preexcitation	↑Creatine kinase ↑Transaminase Lactic acidosis	Global hypokinesia (with/without LV dilatation)	Frequent; large amount nonischemic intramural pattern mostly basal LV inferolateral wall

## Data Availability

Not applicable.

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
