# Peer review of "Personalized Management of Sudden Death Risk in Primary Cardiomyopathies: From Clinical Evaluation and Multimodality Imaging to Ablation and Cardioverter-Defibrillator Implant"

_jpm, 2023, doi:10.3390/jpm13050877_

Round 1
Reviewer 1 Report
General Comments
The authors review a large number of research manuscripts and guidelines to assess the efficacy of personalized management of sudden cardiac death risk in primary cardiomyopathies. They do this by proposing a multi-faceted hierarchical approach that constitutes of 5 steps ranging from clinical evaluations to echocardiographic and echocardiographic analyses, updated ECG monitoring using exercise testing, imaging methods and, following evaluation of risk of SCD, genetic testing to add to a more personalized medicine approach. The authors have provided an easy-to-read collection of brief and summarized components that link to SCD risk and current clinical evaluation of cardiomyopathies. The manuscript is well-written and can reach a broad audience. The nature of this manuscript also provides readers with an overall view of current guidelines to assess risk of SCD.
The authors can improve the manuscript by combining some paragraphs together rather than having stand-alone sentences. In some sections or sub-sections, it would be more beneficial if the authors always related their findings to SCD risk and the importance of reducing risk of SCD within the general population, acknowledging that some subpopulations are more at risk than others (e.g. genetically or in terms of cardiac health). The figures provided might need to be discussed more in text. It would be ideal if the authors guided the readers to what the figures are showing rather than just referring to the figure once with very little context. The authors need to proofread the manuscript for some grammatical errors and sentence structure. In addition, some abbreviations are not previously defined on first use.
Specific Comments:
- In the Abstract, can the authors provide one or two sentences on what their main aim was for this manuscript, acknowledging that it is a review. It would provide the readers with a better understanding of what this manuscript will show.
- Introduction line 33-34: While I appreciate the philosophical statement by Hippocrates, readers might be confused as to what it is referring to (i.e. risk) and would suggest omitting it.
- Introduction line 49: PM is not explained when first abbreviated. I assume it is referring to ‘Personalized Medicine’ or ‘Personalized Management’.
- Section 1.2 line 97: “Sudden death (SD) is mostly due…” I would suggest “Sudden death (SD) occurs mostly due …”
- Section 2 line 131: the word “moment” is too informal. Might need rewording.
- Section 2 line 132-133: not sure what “during effort” is referring to.
- Figure 3: What do the tick marks mean in the figure? The authors can include a brief explanation in the figure legend.
- Figure 3: Are there any specific markers that can be highlighted in Step 2 shown in Figure 3?
- Section 2.1 line 148: “non-considered” should be “not considered”.
- Section 2.1 line 156: The authors can state that NYHA Classification is for heart failure if that is what they are referring to.
- The last paragraph in Section 2.1 might need some rearranging to make it flow better.
- Figure 4 is very poor quality and difficult to read. Can this be changed to a table?
- Section 2.2 line 182: When referring to figure 4, can the authors guide the reader a bit better by discussing which part of the table they are referring to.
- Section 2.2 line 183: “in figure 4” rather than “in the figure 4”
- Section 2.2.1 line 193: The sentence would better read as “... HMC and, while listed as a major risk for SCD”.
- Section 2.2.1 1): Can the authors expand on or give examples of some phenotypic differences of HCM.
- When referring to the HCM Risk-SCD calculator, there is also a research paper that does not seem to be referenced which is quite an important component to the development of this calculator (O'Mahony C, Jichi F, Pavlou M, Monserrat L, Anastasakis A, Rapezzi C, Biagini E, Gimeno JR, Limongelli G, McKenna WJ, Omar RZ, Elliott PM; Hypertrophic Cardiomyopathy Outcomes Investigators. A novel clinical risk prediction model for sudden cardiac death in hypertrophic cardiomyopathy (HCM risk-SCD). Eur Heart J. 2014 Aug 7;35(30):2010-20. doi: 10.1093/eurheartj/eht439. Epub 2013 Oct 14. PMID: 24126876.).
- Section 2.2: Can the authors relate this part of the review back to SCD. It can be a bit unclear on its true relationship.
- Section 2.2.3 line 285: ‘’regurgitation,” – remove the ,
- Section 2.2.3 line 286: LV dyssynchrony; The semi-colon should be a period(.)
- Section 2.3: Can the authors provide some more information on exercise testing, its importance and use in relation to assessment of CMPs, to also allow the general audience to understand its definition.
- Section 2.3 line 300: slight change to the sentence “and HCM, although its utility …”
- Section 2.3 line 303: add a period (.) after the word “events” and start a new sentence with “Moreover”.
- Section 2.3 line 309: same here. Add a period (.) after CMPs and start a new sentence with “In particular, ..”
- When referring to Figure 5, the authors can guide the reader a bit better of what it is showing and what point the authors want to get across to the reader.
- Section 2.4.3: The authors might want to expand a little bit on the usability (including how common) nuclear imaging is in clinical practice.
- Section 2.4.3 line 398: Add period (.) after pattern and start a new sentence with “It has been…”
- Section 2.4.3: The authors can expand, maybe in two or three sentences, on why the highlighted features are related to higher risk of SCD and all-cause mortality.
- Section 2.5 line 427: add “and” before Phospholamban.
- Section 2.5 line 450: add a period (.) after (TPMI) and start the next sentence with “Those associated…”
- Section 2.5 line 456: remove the first instance of “of”
- Section 2.5: Can the authors provide a summary on the link of this hierarchical approach of risk stratification to early detection and precision medicine, in terms of its usability and possible improvement in clinical practice.
- Section 3.1: Can the authors bring back the focus to true SCD events, since the paragraph is quite specific to ACM and while the link to SCD risk can be ‘obvious’, it will flow better if the authors relate their findings back to SCD.
- Section 3.2: The same aspect as the comment above for Section 3.1
- Section 3.3: When referring to Figure 7, can the authors expand a bit more on what the figure is showing. As it stands, it only refers to the last part of the figure, making the rest of the image somewhat redundant. Therefore, the authors can further incorporate a description of the figure in text, to make a point.
- Section 3.3 after line 657: When stating that “there is an urgent need for more precise and patient-tailored risk stratification”, the authors could provide a suitable way forward following their very detailed findings. This could also be part of the conclusion section if it fits better there.
- Section 3.3 line 684: instead of “male carriers so” it could be “male carriers and should be”.
good
Reviewer 2 Report
The authors propose a hierarchical approach for the global estimation of SCD risk in cardiomyopathies. It is already obvious that the historical “LV- EF-centric view” of ischemic heart disease is not sufficient in primary cardiomyopathies. Furthermore, considerations related to interventional treatment are welcomed.
I find the authors' efforts useful for the readers, as they propose a novel perspective supported by hard evidence.
There are too many keywords. Please remove some.
In the abstract, row 20 - 'death'.
Round 2
Reviewer 1 Report
thank you for the revision